# "Air and Visco" Technique: A Promising Innovation in the Surgical Implantation of the Xen Gel Stent Device

Fabrizio Franco [1], Federica Serino [1,2,*] and Fabrizio Giansanti [1,2]

1   Neuromuscular and Sense Organs Department, Eye Clinic, Careggi University Hospital,
    50139 Florence, Italy; francofgs@aou-careggi.toscana.it (F.F.); fabrizio.giansanti@unifi.it (F.G.)
2   Department of Neurosciences, Psychology, Drug Research and Child Health, University of Florence,
    50139 Florence, Italy
*   Correspondence: federica.serino@unifi.it

**Abstract:** We aimed to describe a variation of the surgical technique for the ab interno implantation of the XEN Gel Stent, which, in our experience, is yielding very successful results. The injection of 0.1 mL of air and then of 0.1 mL of a dispersive viscoelastic into the subconjunctival space at the beginning of the surgery allows one to perform a mechanical dissection between the conjunctiva and the Tenon's capsule, creating a real space. In total, 20 eyes of 16 patients underwent the implantation of a stent gel through the "Air and Visco" technique. We retrospectively analyzed the results. We obtained a reduction in the IOP from an average of $18.3 \pm 2.2$ mmHg preoperatively to at $13.5 \pm 3.5$ mmHg at month 12. The needling rate was 20%. We did not register any cases of hypotony (IOP < 6 mmHg), hypotony maculopathy or choroidal detachment. The "Air and Visco" technique allows one to correctly place the device in the subconjunctival space, which the pneumo- and visco-dissection transforms into a real space. This enables an easier surgical performance and more predictable postoperative results, with a low needling rate and reintervention in the follow-up period. It also ensures a greater safety profile because the presence of the OVD on the bleb prevents a sudden lowering of the IOP, eliminating complications such as hypotony, hypotony maculopathy and choroidal detachment in our cohort.

**Keywords:** micro-invasive filtering surgery; glaucoma; bleb management





## 1. Introduction

Glaucoma surgery serves to lower intraocular pressure (IOP) in an effort to prevent future vision loss in glaucomatous patients. Surgery is recommended when medical and laser treatments are ineffective in achieving the target IOP (a level of IOP which is unlikely to cause visual field and optic nerve damage) [1]. Conventional filtering surgery procedures include trabeculectomy and glaucoma drainage devices (e.g., Molteno®, Baerveldt®, and PAUL® implants, Ahmed® Glaucoma Valve, Cucamonga, CA, USA). Recently, new devices have emerged due to their safety profile and lesser invasiveness, termed minimally or micro-invasive glaucoma surgery (MIGS). Several MIGS devices are currently on the market, with different mechanisms of action based on the site of anatomical intervention: (1) Schlemm's canal MIGS devices, where the trabecular meshwork is bypassed, thus directing aqueous humor into the Schlemm's canal; (2) suprachoroidal MIGS devices, which enhance uveoscleral outflow; and (3) subconjunctival MIGS devices, which shunt aqueous humor from the anterior chamber to the subconjunctival space [2].

The Xen Gel Stent (Abbive, Inc., North Chicago, IL, USA) is a recent drainage device developed to lower intraocular pressure in patients affected by primary open-angle glaucoma who are unresponsive to the maximum medical therapy or when surgical treatment has failed. It belongs to the group of subconjunctival MIGS devices; thus, it functions by draining humor aqueous into the subconjunctival space. However, glaucomatous patients

often have inflamed conjunctiva as a result of their topical antiglaucomatous therapy and previous surgeries, and there are commonly adhesions between the two layers [3]. The outcomes of glaucoma surgeries are limited by postoperative fibrosis, and most often, this is due to bleb scarring.

The correct implantation of the XEN Gel Stent is the crucial step for a successful surgery and long-lasting intraocular pressure (IOP) control. It mainly depends on the exact positioning of the stent in the subconjunctival space above the Tenon's capsule; however, the conjunctiva is generally loosely attached to the Tenon's capsule, creating a virtual space. An incorrect positioning of the device, deeper within the Tenon's capsule or embedded in this tissue, is a well-known risk factor for failure: fibrosis may occur with a higher risk of obstruction [4].

The aim of this paper is to describe our experience using a new variation of the standard ab interno surgical technique and our promising results.

## 2. Materials and Methods

This was a monocentric retrospective study on patients diagnosed with primary open-angle glaucoma (POAG), whose IOP was uncontrolled with at least two medications or laser surgery and who were treated with XEN Gel Stent implantations at "Careggi Hospital" Eye Clinic, Univesity of Florence (Italy), from February 2020 to June 2021. The exclusion criteria were a history of any other ocular disorders which would have altered the ocular surface or IOP measurements. Patients lost to follow-up (less than 12 months) were excluded from the data analysis. All patients underwent a complete preoperative visit, which included an IOP measurement (Goldmann Applanation Tonometry-GAT) and dilatated fundus examination (Tropicamide 1%). An assessment of the iridocorneal angle through a gonio lens was performed to verify its wideness and the absence of synechiae and vessels. We also assessed the superior conjunctiva for mobility with a cotton tip.

### 2.1. Surgical Procedure

All surgical interventions were performed by one of the authors (FF). After appropriate disinfection of the periocular skin and conjunctival fornix with povidone iodine 10% and 5%, respectively, anesthesia, as a topical administration of 2.5 mL of lidocaine 2% on the ocular surface, is performed. Then, we proceed to create the "Air and Visco" dissection. The first step is to precisely identify the area of the future bleb, usually in the supero-nasal quadrant; then, we use a dermographic pen and a compass to mark the area at 3.0 mm from the limbus where the Xen Gel Stent will emerge (Figure 1A). After that, we create the bleb: we enter the conjunctiva at 12 o'clock at 3 mm from the limbus with a 27-gauge needle on an insulin syringe and, proceeding upwards with the bevel very superficially beneath the conjunctiva, avoiding the perforation of the Tenon's capsule, we reach previously marked the target area, directing the tip of the needle posteriorly (Figure 1B). At this point, 0.1 mL of air is injected. If the pneumo-dissection is successful, bubbles will form in subconjunctival space (Figure 1C). Then, 0.1 mL of a dispersive viscoelastic (we use ViscoAT, Alcon, Milan, Italy) is injected via the same tunnel (Figure 1D). The bleb is smoothed posteriorly with a cotton tip (Figure 1E). Once the conjunctiva is well separated from the Tenon's capsule, we place the Xen Gel Stent with the traditional ab interno technique. A cohesive viscoelastic (Healon GV Pro, Johnson & Johnson, Irvine, CA, USA) is injected to completely fill the anterior chamber (AC) through a clear corneal incision. The disposable injector enters the AC through the main corneal incision in the infero-temporal sector upon coloring the stent with a blue dye (Trypan blue), and is directed toward the supero-nasal angle. An indirect gonio lens is used to monitor the angle (a direct-view gonio lens can also be used): the correct placement is just anterior to the pigmented trabecular meshwork. Once the needle tip is in the correct place, it is pushed forward through the sclera, emerging in the subconjunctival space 3.0 mm away from the limbus. The injector is actioned, and the needle retracts into the sleeve. The device, correctly positioned, is 1 mm in the AC, 2 mm in the scleral tunnel and 3 mm in the subconjunctival space. The stent is well visible under

the conjunctiva since it has been previously colored, and it must be linear and mobile with the tip (Figure 1F). The correct placement at the angle is verified using a gonio mirror. The viscoelastic is removed from the anterior chamber with balanced salt solutions. Finally, we perform a subconjunctival injection of 0.1 mL of mitomycin C (MMC) 0.02% into the bleb.

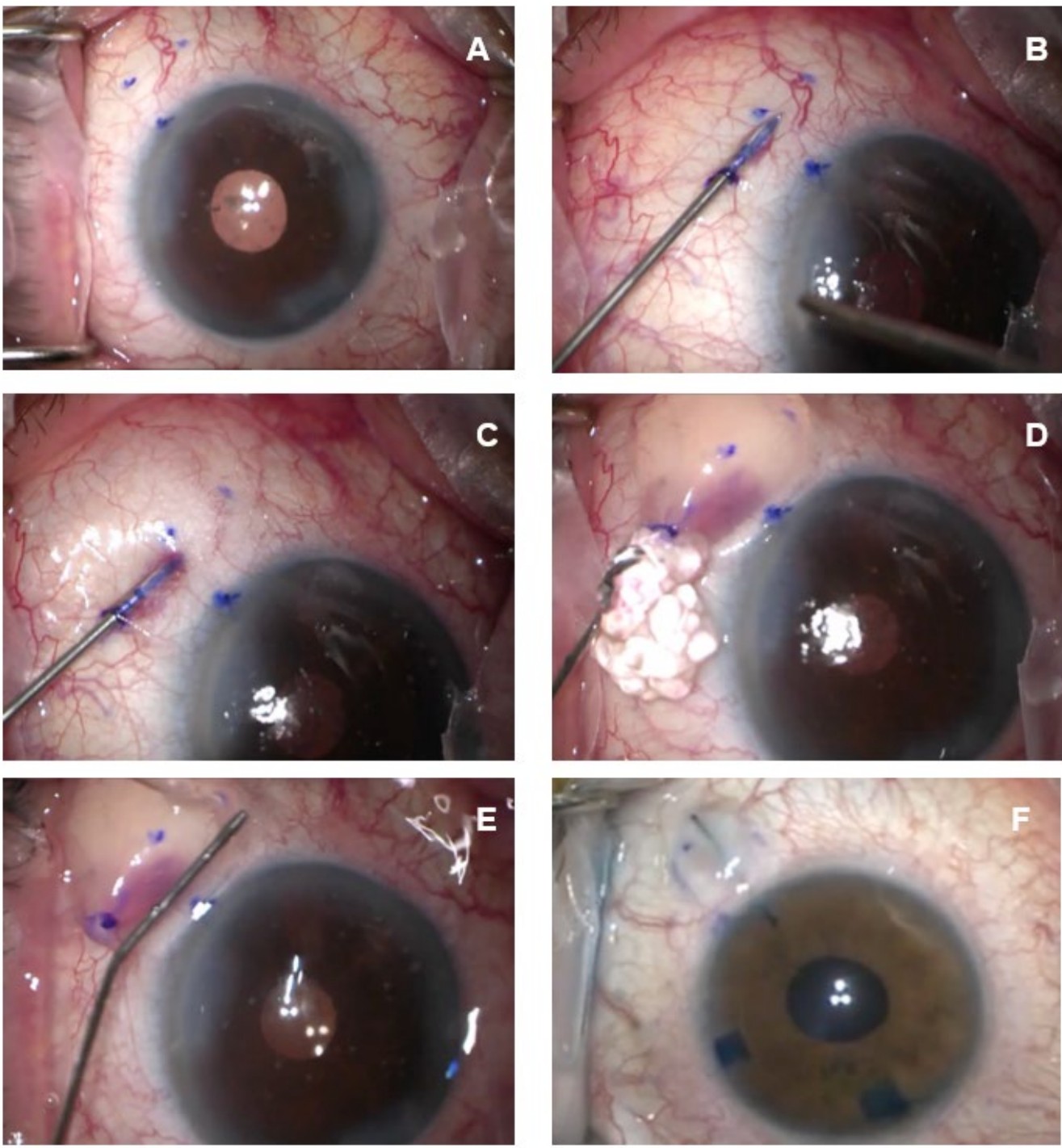

**Figure 1.** (**A**) Once the area of implantation is identified, it is marked with a dermographic pen. (**B**) The 27-gauge needle enters the conjunctiva at 12 o'clock at 3 mm from the limbus. (**C**) The air is injected; if the pneumo-dissection is successful, bubbles will form in subconjunctival space. (**D**) The dispersive viscoelastic (ViscoAT, Alcon) is injected. (**E**) The bleb is smoothed posteriorly. (**F**) The stent is well visible under the conjunctiva.

### *2.2. Postoperative Management*

Patients are instructed to discontinue all glaucoma medications on the day of the surgery. Follow-up visits are conducted at postoperative day 1, every week for the first month and at month 3, month 6 and month 12. Postoperative therapy includes antibiotic prophylaxis for 1 week and steroids, tapered over 3 months. If needed, needling procedures are performed with adjunctive 0.1 mL 5-Fluorouracil (5-FU) on an outpatient basis after the administration of a topical anesthetic (Novesin 0.4% eye drop, Novartis, Milan, Italy).

### 3. Results

We report our preliminary data on 20 eyes of 16 patients, all with a diagnosis of POAG. Six eyes (30%) were already pseudophakic, while 14 eyes were phakic and underwent simultaneous XEN implantation and cataract surgery. Table 1 summarizes the patients' characteristics herein described. The preoperative number of anti-glaucomatous molecules was, on average, $2.5 \pm 0.7$.

**Table 1.** Baseline patients' characteristics.

| Variable | Overall (*n* = 16) |
|:---:|:---:|
| Age (mean value $\pm$ SD) | $76 \pm 7$ |
| Sex (Male/female) | 8/12 |
| **Variable** | **Overall (*n* = 20)** |
| Type of glaucoma: POAG | 20 (100%) |
| Previous surgery: Phaco + IOL | 6 (30%) |
| Number of antiglaucoma medications | $2.5 \pm 0.7$ |

Abbreviations: SD: standard deviation; POAG: primary open angle glaucoma; Phaco + IOL: phacoemulsification + intraocular lens implantation.

We obtained a good reduction in the IOP from an average of $18.3 \pm 2.2$ mmHg preoperatively to $11.3 \pm 2.7$ mmHg 7 days after the surgery. This result persisted throughout the follow-up: at 1 month, the average IOP was $11.2 \pm 2.5$ mmHg ($p < 0.001$); at 3 months, it was $12.4 \pm 1.9$ mmHg ($p < 0.001$); at 6 months, it was $13.6 \pm 3.3$ mmHg ($p < 0.001$); and at 12 months, it was $13.5 \pm 3.5$ mmHg ($p = 0.001$) (Table 2). We used linear mixed models accounting for within-subject correlation to model the change in the IOP over time.

**Table 2.** IOP reduction from baseline.

| Patient | Preoperative IOP | IOP Day 1 | IOP Day 7 | IOP Month 1 | IOP Month 3 | IOP Month 6 | IOP Month 12 | N° of Medications at Month 12 |
|:---:|:---:|:---:|:---:|:---:|:---:|:---:|:---:|:---:|
| 1 | 20 | 10 | 10 | 9 | 13 | 13 | 14 | 0 |
| 2 * | 18 | 10 | 10 | 9 | 11 | 11 | 12 | 0 |
| 3 | 19 | 8 | 10 | 8 | 10 | 19 | 24 | 2 |
| 4 | 15 | 11 | 8 | 9 | 12 | 13 | 14 | 0 |
| 5 | 17 | 12 | 12 | 16 | 16 | 14 | 12 | 0 |
| 6 * | 22 | 8 | 9 | 9 | 10 | 10 | 14 | 0 |
| 7 | 19 | 5 | 6 | 10 | 17 | 16 | NE | NE |
| 8 | 18 | 9 | 8 | 9 | 10 | 11 | 11 | 0 |
| 9 * | 16 | 9 | 14 | 14 | 12 | 11 | 15 | 0 |
| 10 * | 20 | 11 | 11 | 10 | 14 | 17 | 21 | 1 |
| 11 | 15 | 10 | 14 | 12 | 13 | 15 | 11 | 1 |
| 12 | 16 | 8 | 10 | 10 | 12 | 11 | 13 | 0 |
| 13 | 20 | 19 | 15 | 14 | 14 | 13 | 11 | 0 |
| 14 * | 17 | 9 | 11 | 12 | 11 | 14 | 13 | 0 |

**Table 2.** *Cont.*

| Patient | Preoperative IOP | IOP Day 1 | IOP Day 7 | IOP Month 1 | IOP Month 3 | IOP Month 6 | IOP Month 12 | N° of Medications at Month 12 |
|---|---|---|---|---|---|---|---|---|
| 15 * | 18 | 11 | 14 | 12 | 11 | 12 | 12 | 0 |
| 16 | 15 | 18 | 17 | 12 | 11 | 11 | 11 | 0 |
| 17 | 19 | 9 | 12 | 12 | 12 | 12 | 13 | 0 |
| 18 | 21 | 7 | 11 | 10 | 11 | 13 | 12 | 0 |
| 19 | 20 | 10 | 11 | 11 | 12 | 13 | 11 | 0 |
| 20 | 21 | 11 | 13 | 17 | 16 | 24 | NE | NE |
| Mean ± SD | 18.3 ± 2.2 | 10.2 ± 3.2 | 11.3 ± 2.7 | 11.25 ± 2.5 | 12.4 ± 1.9 | 13.65 ± 3.3 | 13.5 ± 3.5 | 0.22 ± 0.55 |

Notes: Patient 3 needed reintervention at month 12, patient 7 needed reintervention at month 8, and patient 20 needed reintervention at month 6. These patients underwent a successful trabeculectomy. The solo procedures are indicated with an asterisk: patients were already pseudophakic. Abbreviations: NE, not estimable.

Complete success was considered as IOP ≤ 18 mmHg at month 12 without the need for medications, surgical revision of the bleb or reoperation [5], for which the rate was 70%. Partial success was defined as IOP ≤ 18 mmHg at month 12 with the use of medications or following bleb revision, for which the rate was 15%. Finally, failure was interpreted as IOP > 18 mmHg with the need for reoperation, for which the rate was 15% (Table 3).

**Table 3.** Surgical success: number of patients (percentage) in each group.

| Variable | Overall (*n* = 20) |
|---|---|
| Complete success, *n* (%) | 14 (70%) |
| Partial success, *n* (%) | 3 (15%) |
| Failure, *n* (%) | 3 (15%) |

Abbreviations: *n*: number.

The needling rate was 20%. Three patients (15%) needed reintervention because of an uncontrolled IOP; two of them had previously received a needling procedure which was ineffective. Two of these patients were already pseudophakic and had previously had an XEN implanted in a solo procedure, whereas one patient had undergone a combined Phaco-Xen surgery. They all underwent a successful ab externo trabeculectomy.

We did not register any cases of hypotony (IOP < 6 mmHg), hypotony maculopathy or choroidal detachment. Two patients (10%) had transitory IOP spikes within the first postoperative week (IOP < 20 mmHg), which resolved without therapy.

The best-corrected visual acuity improved after the combined phacoemulsification-XEN surgery and remained unchanged in the stand-alone procedure.

During the follow-up, we reintroduced antiglaucomatous therapy for three eyes (15%): two of them had a controlled IOP with one molecule, and one required two different molecules. The postoperative number of anti-glaucomatous molecules was, on average, 0.2 ± 0.5.

## 4. Discussion

The XEN Gel Stent (Abbvie, Inc., Chicago, IL, USA) is a microinvasive glaucoma device with a lumen diameter of 45 μm, which works by creating a filtration pathway from the anterior chamber to the subconjunctival space. The correct functioning of the filtering bleb is imperative for the intraocular fluid to be drained from the AC and the IOP to be lowered. The XEN Gel Stent was designed to minimize postoperative fibrosis; the 6 mm device is produced from a highly biocompatible material, porcine cross-linked glutaraldehyde, which reduces foreign body reaction. Furthermore, the ab interno procedure provides for subconjunctival injection without conjunctival peritomy, and this has been demonstrated to involve minimal tissue disruption. However, despite the intraoperative application of antifibrotics, scar tissue often develops. Previous studies report variable needling rates from 10 to 60% [6].

Within this paper, we highlight that mechanical dissection performed through the subconjunctival injection of air and a dispersive viscoelastic in an early phase of the surgical procedure allows one to force the pre-existing adhesion between the conjunctiva and Tenon's capsule. In our cohort, the needling rate was 20%.

We have also observed other advantages. Firstly, the creation of the subconjunctival space, as a real one, permits an easier implantation of the device. In our experience, all implants were well positioned and straight with the tip mobile, and in no case were they excessively curled or requiring manipulation. This eliminated the need for intraoperative needling and, consequently, the risk of subconjunctival bleeding, which can also lead to unwanted early scarring. Intraoperative bleeding and the resulting blood clotting induce the secretion of pro-inflammatory and pro-angiogenic cytokines, especially vascular endothelial growth factor (VEGF), which has a key role in inflammation and postoperative fibrosis [7,8].

We can also assume that the viscoelastic in the bleb reduces the early flow of inflammatory cytokines such as TGFβ, being higher in glaucomatous aqueous humor. This could be one of the variables that lowered the needling rate to 20% in our cohort. Previous studies on other drainage devices seemed to confirm this finding [9,10]. Secondly, we assume that the presence of the viscoelastic in the bleb ensures a more gradual lowering of IOP, minimizing the risk of postoperative hypotony. A low IOP in the first postoperative week is a known complication of filtering surgery and a significant risk factor for choroidal detachment. The rate of CD after trabeculectomy ranges from 2.8% [11] to 18.8% [12].

In fact, the XEN Gel Stent was designed to limit hypotony, lowering IOP safely in a predictable manner based on the Hagen-Poiseuille equation. This law postulates that the pressure differential across a tube with constant dimensions is proportional to the resistance to flow, and this is directly proportional to the length but inversely proportional to the radius of the tube to the fourth power. A tube with a length of 6 mm and 45 μm lumen diameter at an average aqueous humor production of 2–3 μL/min provides a theoretical pressure drop of 8 mmHg, which prevents hypotony [13]. This is not always the case, however. Previous studies reported rates of choroidal detachment which range from 1.4% [14] to 19.8% [15]. There are different explanations: the direct toxicity of MMC to the ciliary body; the long-term use of topical antiglaucomatous therapy (prostaglandins, in particular) with the permanent alteration of the uveoscleral outflow pathway; or an amount of aqueous humor flowing around the tube during the implantation [15].

In our cohort, we did not register any cases of hypotony, hypotony maculopathy or choroidal detachment. No patient required intracameral viscoelastic injection to manage a shallow anterior chamber. We assumed that the time during which the viscoelastic is reabsorbed allows the intraocular inflammation to reduce, but this should be investigated.

By contrast, we observed early transient IOP spikes, which resolved spontaneously within the first postoperative week or resolved with ocular massages, which probably promoted the reabsorption of the viscoelastic. No patient required early topical medication.

## 5. Conclusions

In conclusion, we propose a variation of the standard technique, which is easy to perform, repeatable, more efficient and safer, without an increase in costs. Dissection with air and viscoelastic allows one to make the subconjunctival space a real space, increasing the chance of correcting the positioning the device, reducing the intraoperative manipulation of the tissues, and thus improving the surgical outcomes. The limits of this study are its retrospective design and small sample size.

**Author Contributions:** Conceptualization, F.F., F.S. and F.G.; methodology, F.F. and F.S.; software, F.F. and F.S.; validation, F.F., F.S. and F.G., formal analysis, F.F., F.S. and F.G.; investigation, F.F., F.S. and F.G.; resources, F.F., F.S. and F.G.; data curation, F.F., F.S. and F.G.; writing—original draft preparation, F.F. and F.S.; writing—review and editing, F.F., F.S. and F.G.; visualization, F.F., F.S. and F.G.; supervision, F.G.; project administration, F.G. All authors have read and agreed to the published version of the manuscript.

**Funding:** This research received no external funding.

**Institutional Review Board Statement:** This study was conducted in accordance with the Declaration of Helsinki and approved by the Institutional Review Board of Careggi University Hospital (Regional Ethic Committee for Clinical Trial) (24182_oss).

**Informed Consent Statement:** Informed consent was obtained from all subjects involved in the study.

**Data Availability Statement:** The data presented in this study are available on request from the corresponding author. The data (original imaging) are not publicly available due to privacy issues.

**Conflicts of Interest:** The authors declare no conflict of interest. The funders had no role in the design of the study; in the collection, analyses, or interpretation of data; in the writing of the manuscript, or in the decision to publish the results.

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
