# Peer review of "“Air and Visco” Technique: A Promising Innovation in the Surgical Implantation of the Xen Gel Stent Device"

_2411-5150, 2023_

Round 1

Reviewer 1 Report

Comments and Suggestions for Authors

“Air and Visco” Technique: A Promising Innovation in the Surgical Implantation of the Xen Gel Stent Device

It provides surgical technique for XEN, which is promising method with enhance technique. It is interesting paper with minor suggestion.

Major concern

1.     Please support references for reduce arbitrary description, ex) success of surgery definition? Although XEN is new designed tool, but it is related with bleb management for trabeculectomy.

(Graefes Arch Clin Exp Ophthalmol. 2019 Oct;257(10):2239-2255. doi: 10.1007/s00417-019-04412-0. Epub 2019 Jul 10.)

2.     Do you use MMC or 5FU for augmentation? Why or why not?

3.     Please, provide a rationale or mechanism to support the advantages of this method and to compare it favorably with trab study. This explanation on bleb failure mechanism would enhance significance of the article!

Minor concerns

1.     Table 2 can be changes as IOP graph with SD and IOP is preoperative IOP.

Table 3 should be revised with more fancy visualization and title looks too simple and strange, which did not provide criteria

Comments on the Quality of English Language

Quality of English language looks good and minor correction is necessary.

Author Response

Thank you very much for the review of our manuscript. We sincerely appreciate all valuable comments and suggestions, which helped us to improve the quality of the article. Our responses are described below in a point-to-point manner.

MAJOR POINTS

  • Speaking of definition of surgical success, there is no consensus in literature or a clear evidence base of presumed thresholds. However, generally the upper limit is considered 21 mmHg for early glaucoma and 18 mmHg for moderate-advanced cases. We added the reference in the manuscript. We would maintain this threshold, unless the reviewer says otherwise.

Rauchegger T, Angermann R, Willeit P, Schmid E, Teuchner B. Two-year outcomes of minimally invasive XEN Gel Stent implantation in primary open-angle and pseudoexfoliation glaucoma. Acta Ophthalmol. 2021 Jun;99(4):369-375. doi: 10.1111/aos.14627. Epub 2020 Sep 30. PMID: 32996702; PMCID: PMC8359400.

Fea AM, Durr GM, Marolo P, Malinverni L, Economou MA, Ahmed I. XEN® Gel Stent: A Comprehensive Review on Its Use as a Treatment Option for Refractory Glaucoma. Clin Ophthalmol. 2020 Jun 30;14:1805-1832. doi: 10.2147/OPTH.S178348. PMID: 32636610; PMCID: PMC7335291.

  • Needling procedures were performed with 5FU since it is less dangerous, in our hospital can be administrated on an outpatient basis (whereas MMC cannot) and there aren’t statistically significant differences. We clarified its use in the manuscript.

Halili A, Kessel L, Subhi Y, Bach-Holm D. Needling after trabeculectomy - does augmentation by anti-metabolites provide better outcomes and is Mitomycin C better than 5-Fluoruracil? A systematic review with network meta-analyses. Acta Ophthalmol. 2020 Nov;98(7):643-653. doi: 10.1111/aos.14452. Epub 2020 Apr 30. PMID: 32352646.

  • Since in our study XEN gel stent was implanted with an ab interno technique, whereas Trab is an ab externo procedure, results from our technique cannot be compared to previous study on Trabs.

MINOR POINTS

  • We changed IOP -> preoperative IOP as you suggested. We reported each IOP value separately in the Table because we aimed to better highlight each trend since the small number of patients.
  • We changed the title.

Reviewer 2 Report

Comments and Suggestions for Authors

Fabrizio Franco et al  from Florece, Italy,  have written a paper entitled "“Air and Visco” Technique: A Promising Innovation in the Surgical Implantation of the Xen Gel Stent Device". Injection of 0.1 ml of air and injection of 0.1 ml of dispersive viscoelastic  in the subconjunctival space at the beginning of the surgery  help to obtain a space under the conjunctiva and thus to help the correct  implantation of a small Xen tube. 

 Their report has some weaknesses. The amount of operated eyes is very  small and the study is retrospective.  Results of only 20 eyes of 16 patients are presented. But the way of presentation is demonstrative:  iop of each eye is presented at different time points. It would be great if the combi operated (xen+phaco)  eyes were indicated separately.

 The main thing of this report is a novel technique. The technique is simple and easy to implement.  Actually it is a a good tip for all glaucoma surgeons!

Comments on the Quality of English Language

-

Author Response

Thank you for your positive comments. We highlighted in the table the combi procedures as you suggested.
